# A stochastic assembly model for Nipah virus revealed by super-resolution microscopy

Qian Liu[1], Lei Chen[1], Hector C. Aguilar[2] & Keng C. Chou [1]

Understanding virus assembly mechanisms is important for developing therapeutic interventions. Nipah virus (NiV) is of interest because of its high mortality rate and efficient human–human transmissions. The current model for most enveloped viruses suggests that matrix proteins (M) recruit attachment glycoproteins (G) and fusion glycoproteins (F) to the assembly site at the plasma membrane. Here we report an assembly model that differs in many aspects from the current one. Examining NiV proteins on the cell plasma membrane using super-resolution microscopy reveals that clusters of F and G are randomly distributed on the plasma membrane regardless of the presence or absence of M. Our data suggests a model in which the M molecules assemble at the plasma membrane to form virus-like particles (VLPs), while the incorporation of F and G into the nascent VLPs is stochastic.

[1] Department of Chemistry, University of British Columbia, Vancouver, BC V6T 1Z1, Canada. [2] Department of Microbiology and Immunology, Cornell University, Ithaca, NY 14853, USA. Correspondence and requests for materials should be addressed to K.C.C. (email: kcchou@chem.ubc.ca)

Many virus families, such as the togaviruses, the rhabdoviruses, the para- and orthomyxoviruses, and the retroviruses, undergo assembly at the plasma membrane[1]. For example, extensive electron microscopic and western blot analyses have been carried out for the paramyxoviruses and suggest that the matrix proteins recruit the nucleocapsid and glycoproteins to the assembly sites[2–8]. However, neither electron microscopic studies nor western blot analyses are done on intact cells[9]. Recent breakthroughs in optical super-resolution microscopy have improved the resolution from about 200 nm to as low as 10 nm[10–12], so it is now possible to visualize the organization of the viral proteins at the plasma membrane. An important question in the assembly of enveloped viruses is how the key components for producing a virion are organized and assembled on the plasma membrane.

Nipah virus (NiV) is a member of the *Paramyxoviridae* family[13]. It is an emerging zoonotic virus that causes severe diseases in both animals and humans[14]. Recent NiV outbreaks in southeast Asia have a 40–90% mortality rate[14]. NiV infection can cause fatal encephalitis with a pathological hallmark of endothelial cell–cell fusion[13]. It has been proposed that the NiV matrix proteins (M) recruit the attachment glycoproteins (G) and fusion glycoproteins (F) to the assembly sites either by direct interactions[7] or by co-targeting the same domain at the plasma membrane[15,16].

Here we report a new model for NiV assembly using single-molecule localization microscopy (SMLM). Our images show that small clusters of F and G are randomly distributed on the plasma membrane regardless of the presence or absence of M. The assembly process of NiV virus-like particles (VLPs) is not associated with higher concentrations of F or G co-localizing with the clusters of M. Our data suggests a model in which the M molecules assemble at the plasma membrane to form VLPs while the incorporation of F and G into the nascent VLPs is stochastic. This model predicts that the amount of F and G on the VLPs can be manipulated by controlling the expression levels of F and G in the host cell. We have analyzed 10,000 VLPs expressing both F and G

at different times and confirmed that the amount of F and G on VLPs increases with the expression levels of F and G rather than requiring a fixed stoichiometry dependent on M.

## Results

**NiV-M clusters organize into dome-like structures.** The intracellular protein NiV-M underneath the plasma membrane is thought to direct the assembly and budding process of NiV VLPs[7]. It is known that the expression of M alone is sufficient for the production of NiV VLPs[4]. To study the organization of M, pig kidney fibroblast cells (PK13) expressing NiV-M with green fluorescent protein (GFP) at the N-terminus were fixed, permeabilized, and immunostained with Alexa Fluor 647. Three-dimensional (3D) fluorophore localizations were carried out using a home-built SMLM with dual focal planes which allows an imaging depth of several microns while using the fiducial markers on the coverslip for real-time 3D drift correction[17]. Forty thousand images were acquired at 45 Hz to reconstruct a SMLM image. During image acquisition, sample drift was controlled within 1 nm (root mean square, RMS) in the lateral directions and 3 nm in the axial direction[17]. The fluorophore localization precisions were ~10 nm (standard deviation, SD) laterally and 30 nm axially[17,18]. Images taken through the middle of the cell (position 1 in Fig. 1a) show distinct M puncta with a diameter of 20–40 nm inside the cell (Fig. 1b). However, images at the plasma membrane (position 2 in Fig. 1a) reveal clusters with diameters of a few hundred nanometers (Fig. 1c). Our findings provide in situ evidence showing that M proteins further assemble at the plasma membrane, which is consistent with the self-assembly model proposed for the human metapneumovirus M in the presence of 1,2-dioleoyl-sn-glycero-3-phosphocholine in solution[19].

We observed that some M clusters were organized into dome-like structures at the plasma membrane. An example is shown in the boxed region in Fig. 1d on the dorsal surface of the cell (position 3 in Fig. 1a). These dome-like structures could not be observed at the ventral surface of the cell in contact with the

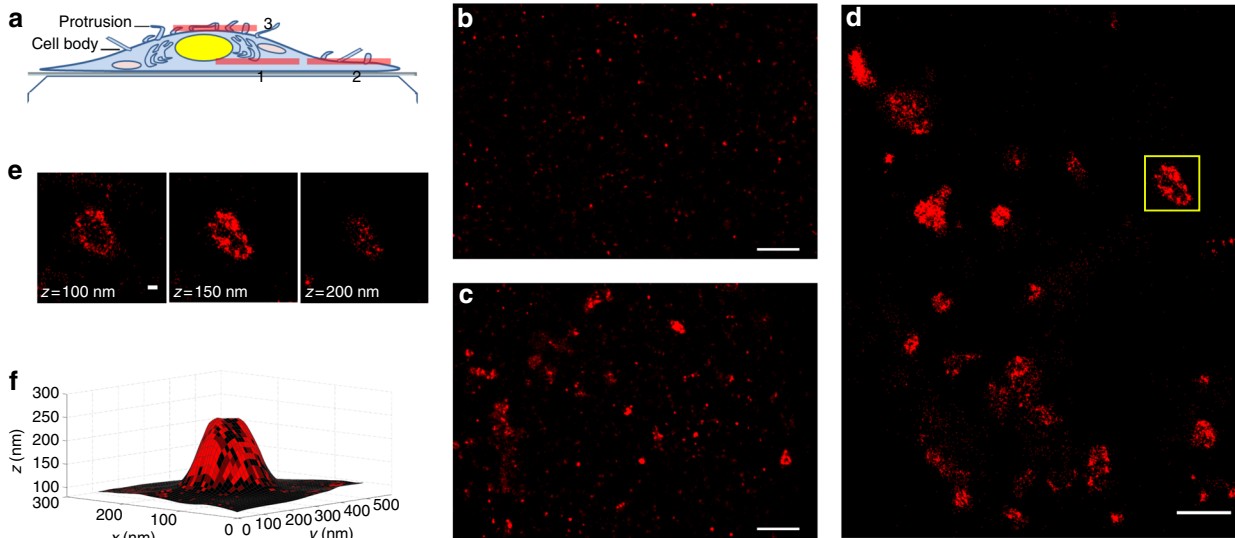

**Fig. 1** Distribution of NiV-M in PK13 cells. PK13 cells expressing NiV-M were fixed, permeabilized, and labeled with goat anti-GFP and anti-goat Alexa Fluor 647 antibodies. **a** Schematic illustration of the imaging planes of a cell. **b** *x–y* cross section (100 nm thick in *z*) of a region through the middle (position 1 in **a**) of a representative cell showing the M clusters. Scale bar: 1 μm. **c** *x–y* cross section (100 nm thick in *z*) of a region at the plasma membrane (position 2 in **a**) of a representative cell showing larger M clusters. Scale bar: 1 μm. **d** *x–y* cross section of a region (position 3 in **a**) of a cell showing a representative dome-like structure formed by M. Scale bar: 1 μm. **e** *z*-stacks of the *x–y* cross section of the dome-like structure boxed in **d**. Scale bar: 0.1 μm. **f** 3D surface reconstructed by using the M localizations in **e**, with M localization density projected on the 3D surface. A higher brightness indicates a higher localization density. One representative cell image out of three independent experiments (*n* ≥ 30) is shown

coverslip. The z-stacks of the dome-like structure are shown in Fig. 1e, and the 3D surface reconstructed by using the M localizations is shown in Fig. 1f. The organization of M on these dome-like structures was fragmented, not continuous. This is in agreement with the recent cryo-electron microscopy studies showing fragmented M patches in the virions of Newcastle Disease Virus[20].

**NiV glycoproteins form clusters on the plasma membrane**. To investigate the organization of the viral envelope glycoproteins on the plasma membrane, we used FLAG-tagged F[21] and hemagglutinin (HA)-tagged G[22] constructs (Supplementary Fig. 1). PK13 cells expressing either F or G were fixed at 24 h post transfection, immunostained for F or G via tags, and imaged at the dorsal surface of the cell without permeabilization (position 3 in Fig. 1a). F formed distinct clusters (Fig. 2a) compared to the more dispersed clusters formed by G (Fig. 2b). Since both F and G were abundant on the plasma membrane, the localizations delineated the entire plasma membrane (Fig. 2a, b). The observed membrane structures resembled those imaged by scanning electron microscopy (Supplementary Fig. 2). Both F (Fig. 2a) and G (Fig. 2b) were detected on the cell body and membrane protrusions, which are shown as tubular structures with a relatively uniform diameter of ~200 nm. To determine whether the localization densities of F or G were different between the cell body and membrane protrusions, we sampled a total of ~20 areas from each type of region and compared their localization densities. To account for the cell-to-cell variation in the expression levels of F and G, the localization densities on the protrusions were normalized against the average density of the cell body. We found that the localization densities of F or G on membrane protrusions were not statistically different from those on the cell body (Fig. 2c). A similar conclusion was obtained for cells fixed at 16 h post transfection (Supplementary Fig. 3c). Therefore, our data show that the distribution of the viral envelope glycoproteins is generally uniform over the plasma membrane.

F generally exhibited a greater Hopkins' index than G (Fig. 2d), suggesting more extensive clustering of F than G at the plasma membrane[23]. This observation is similar to the recent SMLM study on HIV Env and Influenza hemagglutinin, both of which demonstrate clustering behavior at the plasma membrane[24,25]. Similar distribution patterns of the envelope glycoproteins were obtained on the plasma membrane of PK13 cells at a shorter expression period (16 h). This suggests that the distribution and arrangement of F and G are not significantly dependent on the cell surface expression levels (Supplementary Fig. 3).

To determine the co-localization of F and G, we acquired dual-color SMLM images of F and G at the plasma membrane (Fig. 2e). A numerical co-localization analysis was carried out using the coordinate-based co-localization algorithm previously developed by Malkusch et al[26]. The degree of co-localization (DoC) is calculated for every single-molecule localization and has a value from −1 for segregation, through 0 for random distributions, to 1 for a high probability of correlated co-localization[26–28]. A random distribution indicates that the co-localization of two molecules is a random event rather than regulated by a specific mechanism. In this algorithm, the DoC value is dependent on the maxima radius ($R_{max}$) used for the DoC analysis[26,27]. A $R_{max}$ of 100 nm was used in this study to reflect the size of a typical VLP. The effect of the selected $R_{max}$ on the DoC values is demonstrated in the Supplementary Fig. 4. The analysis indicates that F and G are mostly localized in segregated clusters as the DoC values show a maximum in the negative range (Fig. 2f). The analysis partially explain that the association of the glycoproteins observed by co-immunoprecipitation assays can be

due to random events[22,29,30]. Furthermore, we found that the coexistence of these two NiV envelope glycoproteins did not affect each other's clustering behavior (Fig. 2e): F formed distinct clusters and G formed more dispersed clusters. PK13 cells are non-permissive for NiV entry because they lack the NiV receptors ephrinB2/B3[13]. Nonetheless, similar behaviors were also observed on NiV permissive HeLa cells, which expressed the ephrinB2/B3 receptors[13] (Supplementary Fig. 5). Therefore, the organization of F and G is independent of the presence of the receptors.

**NiV-M does not alter the distribution of glycoproteins**. To investigate whether M actively recruits F and G at the plasma membrane, three-color images were collected for cells simultaneously expressing M, F, and G. Both F and G were imaged using SMLM, while diffraction-limited images of M-GFP were used to identify regions with large M clusters. Figure 3a shows the F (red), G (green), and M (blue) clusters at the plasma membrane of the cell body (position 3, Fig. 1a). At a higher z position above the cell body, the membrane protrusions could be seen (Fig. 3b). We observed that the clusters of F and G were situated on dome-like structures (1 and 2 in Fig. 3b, c) in the M-positive regions. The shape of the dome-like structures recapitulated those formed by M localizations when expressed alone in the cell (Fig. 1e, f). Figure 3d shows the 3D reconstructed surface using the localizations of F and G in region 1 of Fig. 3b. These dome-like structures of G and F could not be observed when M was absent (Fig. 2a, b). Therefore, it is plausible that these dome-like structures formed by M are the assembly sites of the VLPs.

Interestingly, the localization densities of F and G at the M-positive regions were not statistically different from those at the M-negative regions (Fig. 3e). Additionally, the Hopkins' indices (Fig. 3f) and DoC values (Fig. 3g) of F and G were comparable in the M-positive and M-negative regions. Furthermore, dual-color SMLM images indicate non-correlated distributions among M, F, and G with negative averaged DoC values (Fig. 2f and Supplementary Fig. 6). These observations disagree with the commonly believed model that the viral envelope glycoproteins coalesce to the matrix protein for the assembly of the nascent virions[7,8,31,32]. Previous studies using western blot analysis suggest that F may facilitate G to incorporate into VLPs[7]. Nonetheless, we did not observe a significant difference on the DoC values between M and G with or without the presence of F (Supplementary Fig. 6c, d and e). Figure 3h shows the z-stacks of a representative VLP, marked by the GFP signal from M (blue). We found that the spatial organization and distribution of F and G on the VLP membrane were similar to that of the host cell's plasma membrane (Fig. 2e and 3a–c). This observation indicates that the clusters of the envelope glycoproteins on the plasma membrane have not been considerably rearranged when incorporated into the VLPs. All together, these findings suggest an alternative assembly model for NiV, in which no active recruitment of the envelope glycoproteins to M is involved. Instead, the incorporation of the envelope glycoproteins occurs stochastically upon the envelopment of the M assemblies at the host cell's plasma membrane.

**Incorporation of the glycoproteins into VLPs is stochastic**. If the incorporation of the envelope glycoproteins into VLPs occurs stochastically, the model predicts that the amounts of the envelope glycoproteins in VLPs should correlate with their expression levels on the host cell membrane rather than showing a fixed stoichiometry of G/M or F/M. To test this model, we collected images of ~10,000 VLPs at 18 and 45 h post transfection of the viral envelope glycoproteins and analyzed the intensity of M, F,

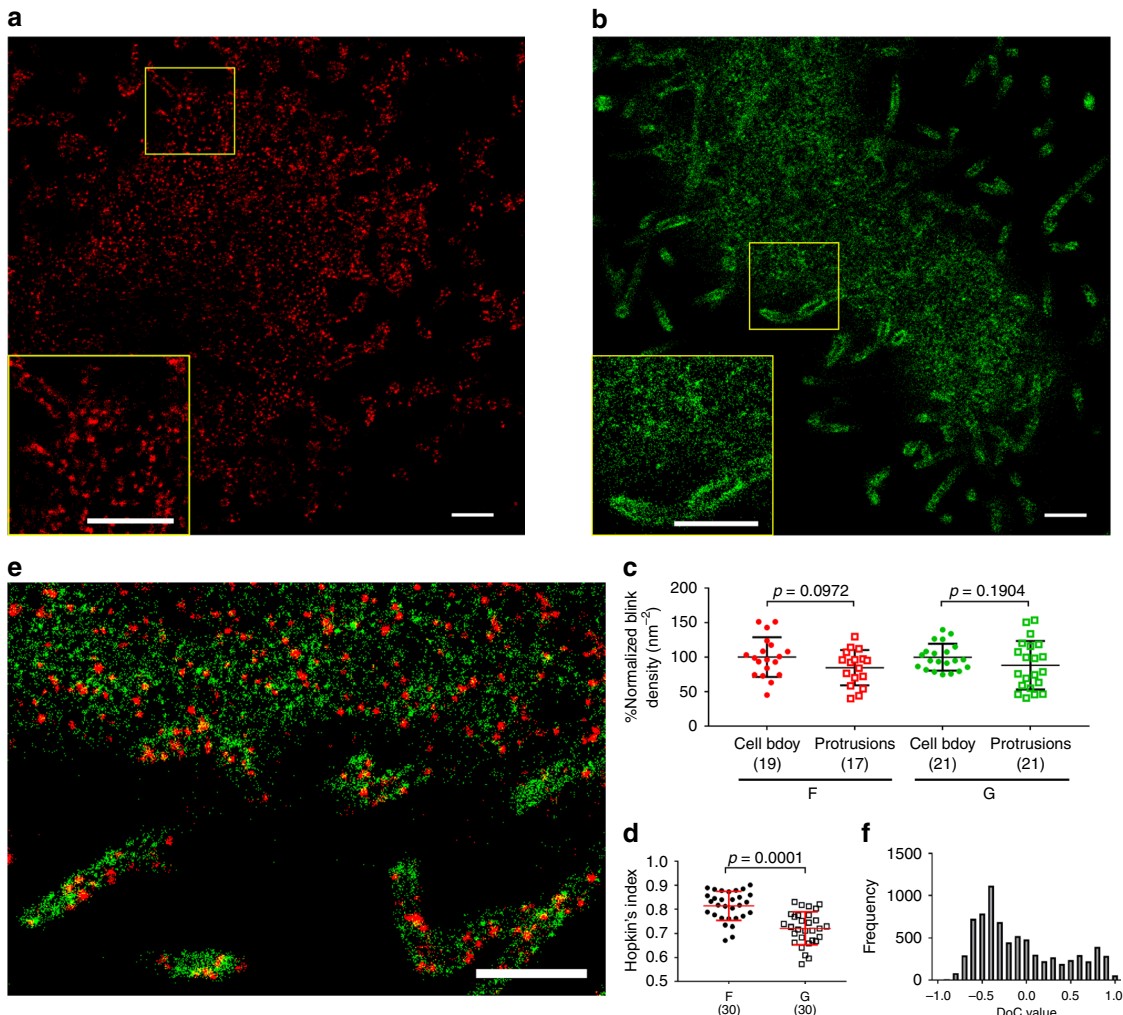

**Fig. 2** Distribution of NiV envelope glycoproteins on the plasma membrane of PK13 cells. PK13 cells expressing NiV-F or/and -G were fixed at 24 h post transfection and immunostained. Without permeabilization, NiV-F was immunostained by using a mouse anti-FLAG primary antibody and -G a rabbit anti-HA primary antibody. For single-color SMLM, Alexa Fluor 647 secondary antibodies were used for detection. For dual-color SMLM, Alexa Fluor 647 secondary antibodies were used for detection of F, and Cy3B secondary antibodies for G. **a**, **b** x–y cross section (100 nm thick in z) at the dorsal surface (position 3 in Fig. 1a) of a representative cell expressing F (**a**) or G (**b**). The boxed region is enlarged to show the detailed distribution pattern. **c** Comparison analyses of the localization densities of the F (red) or G (green) at the cell body versus membrane protrusions. Each data point was calculated using an area of 0.2 × 0.2 μm². All data were normalized to the mean of the cell body. **d** Hopkins' index of the F and G localizations from $n = 30$ cells. Lines represent the mean value and SD. The sample size is indicated in the parentheses. The p values were determined by two-tailed, unpaired t-test with Welch correction. **e** x–y cross section (100 nm thick in z) of a region at the dorsal surface (position 3 in Fig. 1a) of a representative cell co-expressing F (red) and G (green) with a pixel size of 10 nm. Scale bars: 1 μm. **f** The distribution of the DoC values between F and G molecules. One representative cell image out of three independent experiments ($n ≥ 30$) is shown

and G on VLPs. Fig. 4a, b shows the intensity distributions of F and G in the VLPs, respectively. Both F and G showed significantly higher intensity at 45 h than those at 18 h post transfection, which was consistent with the cell surface expression levels of F and G on the host cells measured by flow cytometry (Fig. 4e). Moreover, the VLPs collected at 45 h post transfection also showed higher intensity ratios for both G/M and F/M. This observation indicates the stoichiometries of G/M and F/M in the NiV VLPs vary with the expression level on the host cells (Fig. 4c, d). These results confirm that the incorporation levels of the envelope glycoproteins are highly dependent on their expression levels at the host cell membrane and may not be regulated by M.

## Discussion
Our images suggest that the co-localization of the F and G clusters is random. Since NiV-induced membrane fusion requires

F and G to act in concert[22], our finding indicates that a specific spatial organization of F and G may not be needed to trigger membrane fusion. Although previous analyses using the co-immunoprecipitation assay[29,30] and flow cytometry-based protein interaction assay[21] have detected interactions between F and G proteins, these methods do not provide information on whether these interactions are non-random or even occur in intact cells.

Since NiV is a biosafety level-4 agent, most models of the NiV life cycle have been proposed using plasmid transfection-based methods[33]. Whether these models can be applied to the real virus remains an open question. Nonetheless, our studies reveal a virus assembly model previously unrecognized by electron microscopy and biochemical analyses. Our findings suggest that the assembly process of some enveloped viruses could be much simpler than previously envisioned.

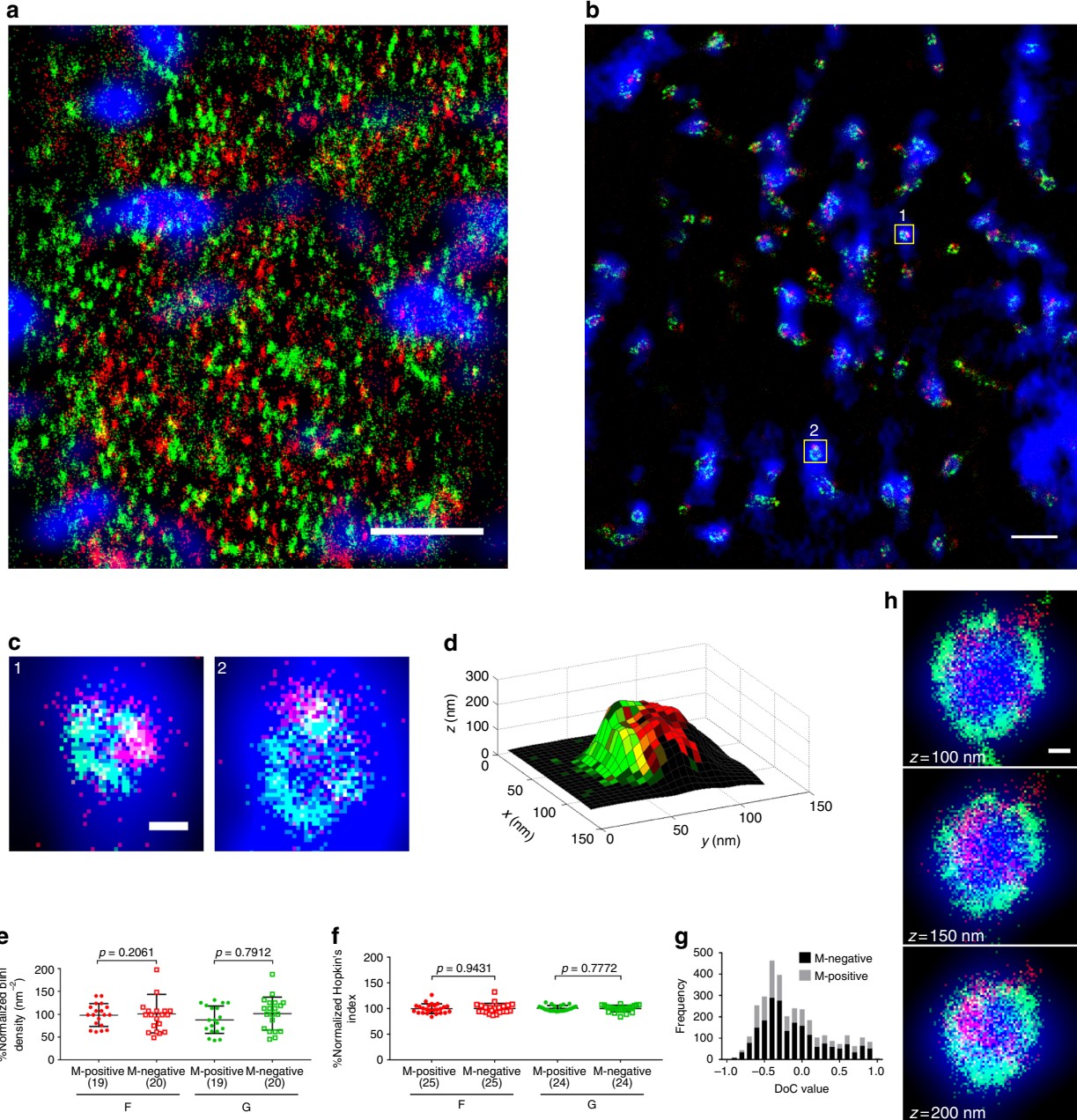

**Fig. 3** Presence of M does not affect the arrangement of the envelope glycoproteins' clusters on the membrane. PK13 cells were co-transfected with NiV-M, -F, and -G and fixed at 24 h post transfection. Without permeabilization, F was immunostained using a mouse anti-FLAG primary antibody and an anti-mouse Cy3B secondary antibody, and G an anti-HA primary antibody and an anti-rabbit Alexa Fluor 647 secondary antibody. **a**, **b** $x$–$y$ cross section (100 nm thick in $z$) of a representative cell shows the superimposition of a wide field image for M (blue) and the corresponding SMLM images of F (red) and G (green) on the cell body (**a**) and membrane protrusions (**b**). Scale bar: 1 μm. **c** $x$–$y$ cross section (100 nm thick in $z$) of the M-positive sites (**c1**, **2**) boxed in **b**. Scale bar: 0.1 μm. **d** 3D surface reconstruction of the dome-like structure in **c1**, with F (red) and G (green) localization densities projected on the surface. A higher brightness indicates a higher localization density. **e** Comparison of the localization densities of F (red) or G (green) on the M-positive and M-negative regions at the dorsal surface of the cell. **f** Comparison of the Hopkins' indices of F (red) and G (green) in the M-positive and M-negative regions at the plasma membrane. All data in **e** and **f** were normalized to the mean of the M-negative regions of the same cell. Lines represent the mean value and SD. The sample size is indicated in the parentheses. The p values were determined by two-tailed, unpaired $t$-test with Welch correction. **g** The distribution of the DoC values between F and G molecules in the M-negative and M-positive regions. **h** VLPs produced in PK13 cells expressing M, F, and G were adhered to fibronectin-coated coverslips, fixed, and stained for NiV-F and G via tags described above. The $z$-stacks of the $x$–$y$ cross section of a VLP show the super-imposition of a wide field image of M (blue) and the corresponding SMLM images of F (red) and G (green). Scale bar: 0.1 μm

## Methods

**Cell culture**. PK13 (pig kidney fibroblast, American Type Culture Collection; ATCC, CRL6489)[13,34] and HeLa cells (ATCC, CCL-2) were grown at 37 °C and 5% $CO_2$ in Dulbecco's modified Eagle's medium supplemented with 10% fetal bovine serum, 50 IU penicillin per ml, 50 μg streptomycin per ml, and 2 mM glutamine (Life Technologies). Cell cultures were monitored routinely for mycoplasma

contamination by using a mycoplasma detection PCR kit (Applied Biological Materials).

**Expression plasmids and transfection**. The codon-optimized NiV-G with a C-terminal HA tag was constructed as previously described (genbank accession no.

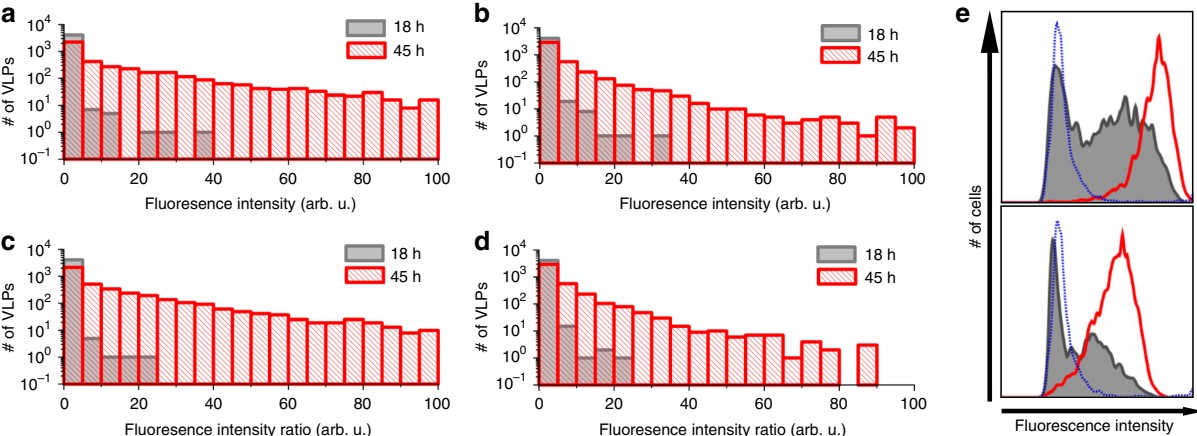

**Fig. 4** The incorporation of the envelope glycoproteins is related to the cell surface expression levels, not the stoichiometry of F/M or G/M. VLPs described in Fig. 3h were subjected to confocal microscopy imaging. The VLPs were located by the GFP signal from M; the intensities of the F and G signals co-localized with M were analyzed. **a, b** Histograms of F (**a**) and G (**b**) incorporation levels in VLPs collected at 18 (gray) and 45 (red) h post transfection of F and G. **c, d** Histograms of F/M (**c**) and G/M (**d**) ratios in VLPs collected at 18 (gray) and 45 (red) h post transfection of F and G. **e** Expression levels of F (top) and G (bottom) in the GFP-positive and VLP-producing cells at 18 (gray) and 45 (red) h post tranfection. Cells expressing M alone were set as control (blue dotted line). Three independent experiments were performed

AY816745.1)[29]. NiV-M-GFP was generated by fusing enhanced GFP sequence in frame to the N-terminus of a codon-optimized NiV-M construct (genbank accession no. EU480491.1). The VLP assembly and budding ability of this NiV-M-GFP construct is comparable to the untagged NiV-M[35]. A FLAG tag was inserted after residue 104 of the codon-optimized NiV-F (genbank accession no. AY816748.1). The FLAG-tagged F construct is functional in a cell–cell fusion assay shown in Supplementary Fig. 1 and by Stone et al.[21]. All constructs were cloned into the pcDNA 3.1+ (Invitrogen, V79020) expression plasmid. PK13 or HeLa cells were seeded on coverslips (18 mm, 1.5 H, Marienfeld) coated with 2.5 µg per well fibronectin (Sigma-Aldrich) in a 12-well plate, and transfected with 1 µg of plasmids per well using Turbofect (Thermofisher Scientific) on the next day.

**Antibodies and immunofluorescence**. At 16 or 24 h post transfection, cells were fixed with phosphate-buffered saline (PBS) containing 4% paraformaldehyde (PFA) and 0.2% glutaraldehyde for 90 min (Electron Microscopy Sciences)[36]. Plasma membranes were permeabilized with 0.1% Triton X-100 (Sigma-Aldrich) for the detection of the GFP-tagged NiV-M in SMLM. Cells were incubated in signal enhancer Image-IT-Fx (Life Technologies) for 45 min, and then blocked using BlockAid (Life Technologies) blocking solution for 1 h at room temperature. The HA.11 rabbit polyclonal antibody (Biolegend, 902301) was used at a 1:900 dilution to detect the HA-tagged NiV-G and the M2 mouse monoclonal anti-FLAG antibody (Sigma-Aldrich, F1804) was used at a 1:200 dilution to detect the FLAG-tagged NiV-F. NiV-M was detected either by imaging GFP fluorescence or by using a goat anti-GFP antibody at a 1:600 dilution (abcam, ab5450) followed by a secondary antibody. Alexa Fluor 647- (Life Technologies, A21235, A21447, A21244, and A31571) and Cy3B (GE Healthcare, PA63101)-conjugated secondary antibodies were used at 1:300 and 1:100 dilutions, respectively. The Cy3B-conjugated secondary antibodies (Jackson ImmunoResearch, 111-005-008, 711-055-152, and 705-005-147) were manufactured by Ablab (Vancouver, Canada). Cells were incubated with primary antibodies overnight at 4 °C, and then with secondary antibodies for 1 h at room temperature. Each antibody binding step was followed by five washes with PBS. Cells were then fixed in PBS containing 4% PFA for 10 min at room temperature.

**SMLM setup**. Imaging was performed using a home-built microscope with a sample stabilization system. The details of the microscope has been published previously[18]. Briefly, four lasers were used in the excitation path: a 639 nm laser (Genesis MX639, Coherent) for exciting the Alexa Fluor 647; a 532 nm laser (Opus, Laser quantum) for exciting the photo-switchable Cy3B; a 488 nm laser (DHOM-100B, Fine Mechanics) for exciting GFP; and a 405 nm laser (LRD 0405, Laserglow Technologies) for reactivating the Alexa Fluor 647 and Cy3B. All four lasers were coupled into an inverted microscope equipped with an apochromatic TIRF oil-immersion objective lens (60×; numerical aperture 1.49; Nikon). The emission fluorescence was separated using appropriate dichroic mirrors and filters (Semrock)[17], and detected by electron multiplying charge-coupled device cameras (Ixon, Andor). A feedback loop was employed to lock the position of the sample during image acquisition. Sample drift was controlled to be <1 nm laterally and 2.5 nm axially.

**SMLM image acquisition and reconstruction**. Fluorescent beads (F8799, ThermoFisher Scientific) were added to the sample as fiducial markers. Samples were

immersed in oxygen-savaging buffer supplemented with 50 mM mercaptoethyla-mine or 140 mM β-mercaptoethanol during imaging acquisition[37]. The expression level of the protein of interest in individual cells was determined by measuring the average emission fluorescence intensity of an area of 27 × 27 µm$^2$. Cells with an emission fluorescence intensity of threefold or greater than that of the mock-transfected cells were selected for imaging. For SMLM imaging, cells were exposed to a laser power density of 1 kW cm$^{-2}$ for the 639 and 532 nm lasers to activate the Alexa Fluor 647 and Cy3B, respectively. In all, 40 000 images were acquired at 45 Hz to reconstruct each SMLM image. For dual-color SMLM, image acquisition at each channel was performed sequentially. Overlapping of these two colors were carried out using ~40,000 images of fluorescent beads recorded at various positions of these two cameras to find an optimal geometric transformation. The resulting color-overlapping error is ~10 nm (RMS). Custom software written in MATLAB (Mathworks) was used to reconstruct SMLM images.[17]

**VLP production and immunofluorescence**. VLP were manufactured by sequentially transfecting the PK13 cells with the aforementioned NiV-M, -F, and -G encoding plasmids using polyethylenimine (Sigma-Aldrich). Briefly, the NiV-M plasmids were transfected at time 0, and F and G plasmids were transfected at 0 or 27 h post transfection of M. At 45 h post transfection of M, VLPs were collected on a 20% sucrose cushion by centrifugation of the cell supernatant at 100 000 × g. The VLPs were resuspended in 5% sucrose-NTE buffer[38]. VLPs were immobilized on the coverslips (12 mm, Thermofisher Scientific) coated with 1.5 µg fibronectin for 2 h at room temperature, followed by 4% PFA fixation. The procedures for VLP SMLM imaging are exactly the same as that of the cells. For confocal microscopy imaging, VLPs were incubated with BlockAid blocking solution at room temperature for 1 h. NiV-F and G proteins were stained with the primary and secondary antibodies mentioned above, and embedded in prolong diamond antifade mountant (Thermofisher). A 3D Z-scanning protocol was performed to image VLPs on a Zeiss III spinning disk confocal microscopy (Zeiss). The intensities of F and G on VLPs were analyzed using custom software written in MATLAB. The VLPs were marked by GFP signals, and the signals of F and G were analyzed only if they co-localized with the GFP signal. The GFP-positive cells were analyzed by a flow cytometer (FACSARIA III, BD Biosciences) and FLOWJO v10 software.

**Statistical analysis**. The p values of the localization density and the Hopkins' index data sets were determined by two-tailed, unpaired t-tests with the Welch correction. All statistics were performed using GraphPad Prism 7 software. Dot plots include a horizontal line representing the mean value and whiskers representing SD.

**Data availability**. The data and computer codes that support the findings of this study are available from the corresponding author upon request.

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

## Acknowledgements

This work was financially supported by the Natural Sciences and Engineering Research Council of Canada and the Canada Foundation for Innovation. We thank Dr. Linda Matsuuchi and Dr. Michael Gold for providing laboratory space in the Life Sciences Institute at the University of British Columbia. We also thank Dr. David Scriven, Dr. Birgit Bradel-Tretheway, Dr. Bharat Joshi, Dr. Libin Abraham, and Ms. May Dang-Lawson for technical support and helpful discussions. We thank Dr. Benhur Lee at the Icahn School of Medicine at Mount Sinai for providing the NiV-M-GFP expression plasmid.

## Author contributions

Q.L. designed and performed the experiments and analyzed the data. K.C.C. built the SMLM microscope. L.C. and K.C.C. wrote the programs for the localization density and coordinate-based co-localization analyses. H.A.C. provided the NiV-F-FLAG and NiV-G-HA expression plasmids and PK13 cells, as well as data interpretation. Q.L. and K.C.C. wrote the manuscripts with input from H.A.C.

## Additional information

**Competing interests:** The authors declare no competing interests.

