## [Peer Review File · Nature Communications]

Reviewers' comments:

Reviewer #1 (Remarks to the Author):

In the manuscript by Liu et al., dSTORM was utilized to evaluate the localization of the F, G, and M proteins of Nipah virus, expressed from plasmids. Overall, this is an interesting paper, that comes to the conclusion that the M protein does not effect interactions or incorporations of F and G into VLPs, but that this is stochastic. This is an surprising conclusion, fully supported by the data, but it would be more convincing if a number of experiments were added to the manuscript. It should be noted, that at some point in the arc of this research, these experiments really do need to be performed with the entire virus, to validate the findings. I think this should be mentioned in the manuscript.

Major issues

- 1) Is it known whether the GFP inserted into the NiV-M, interferes with normal viral processes? Has this been confirmed. GFP addition to other paramyxovirus M proteins impedes filament formation and normal assembly. Given that working with the whole virus is difficult, comparisons should be made with and without the GFP tag, if an anti-M antibody exists, and if not, then with smaller epitope tags, such as Flag or V5 or HA.
- 2) Based on some of their recent work (Johnston et al., 2017) – it would be very useful to image G and M and F, G, M in order to see if F facilitates F/M colocalization. In the recent paper in JVI, they demonstrate that F facilitates G incorporation into VLPs, therefore seeing if F effects G localization with M structures would make sense.
- 3) If the F-cytoplasmic tail is removed - does it change the associations?
- 4) Is M localized to caveolae? Are F and G in different membrane structures?
- 5) If you remove cholesterol - does this alter the localization of M?
- 6) Where is the actin cortex in these images? Does the cortex deform during M expression? Where are the microtubules relative to these structures?
- 7) Endothelial cells, one of the targets of NiV, are polarized. Does polarization effect the localization of the F, G or M? MDCKs (Lamp et al., JVI, 2013) on transwells could also be used.

Reviewer #2 (Remarks to the Author):

In this study, authors utilise the super-resolution single molecules localisation microscopy (SMLM) approach to investigate the behaviour of Nipah virus surface glycoproteins G and F (responsible for virus attachment and fusion, respectively) at virus assembly sites. Here, authors conclude that the incorporation of the virus surface proteins into newly assembled virus proceeds in stochastic manner that is not modulated by the viral structural protein M. This virus assembly model differs to the virus assembly models proposed in previous studies of viruses such as HIV-1 or influenza where incorporation of virus surface proteins is thought to be dependent on direct or indirect interactions with virus structural proteins.

In general, the manuscript is short, concise and its story is easy to follow and understand. On the other hand, this study suffers from issues which cast doubts on authors' claims and

therefore need to be addressed.

Concerns:

1. In this study authors utilise full antibodies to tag the viral proteins of interest. The multivalent nature of both primary and secondary antibodies used raises concerns that it will induce artificial clustering of the proteins. Such clustering may therefore give a false result with regards to the observed distribution of viral proteins on the virus surface, especially in single molecule resolution SMLM approach. This is especially a concern in the samples that were post-fixed following the antibody labelling (Figure 2 and 3). Furthermore, since even a very long and strong fixation does not complexly immobilise the proteins (Ref: Tanaka et al. 2010 doi:10.1038/nmeth.f.314) antibody-induced protein clustering may also be a problem in pre-fixed samples (Figure 1).

To address this issue authors should utilise Fab fragments for both primary and secondary antibodies for their imaging experiments. At the very least authors should conduct Fab based control experiment that assesses the potential impact of antibody induced protein clustering on the distribution of investigated proteins.

2. In Figure 2 authors claim that lack of significant overlap between proteins F and G, signified by very few yellow "overlap" pixels (Line 88), indicate that these proteins have a limited association (Line 90) in contrast to previous bulk studies (Line 91). However, given the fact that authors themselves state (line 89) that the localization precision of the microscope used is ~10 nm which is roughly the size of a single antibody the co-localisation estimation by yellow "overlap" pixels can be misleading. At such high accuracy and precision levels the size of tagging antibodies themselves will prevent any overlap as no two antibody complexes can occupy the same physical space.

Authors should indicate this caveat more clearly or use another method to assess the protein co-association that have been used in super-resolution SMLM experiments (Ref: Nicovich et al. 2017 doi:10.1038/nprot.2016.166 and Georgieva et al. 2016 doi:10.1016/j.ymeth.2016.03.029)

Furthermore, the use of the relatively large full antibody tags may also cause selective lack of labelling of very closely associated F and G proteins due to the occlusion by the antibody complexes themselves. Such behaviour may result in the perceived lack of association between proteins. This would normally not be a problem in a classic diffraction limited microscopy experiments but becomes a potential pitfall in super-resolution based studies of protein association.

To address this authors should consider performing an experiment with smaller tags such as Fabs or nanobodies.

3. In Figure 3, which provides the data for the main point of this study, authors conclude that since there is no difference in the localization densities of F and G proteins in the areas with or without M protein this indicates that the incorporation of these proteins to the budding viruses may proceed in purely stochastic without any interactions with protein M. With such a strong claim which represents the main conclusion of the paper, it is surprising that the authors did not perform a more in-depth analysis of the data to strengthen their case, especially since it is more difficult to convincingly prove a lack of a process than to

demonstrate that it exists. For example, as with Figure 2 data, authors should consider performing Hopkins index analysis (or other spatial analysis) of F and G protein distributions in the areas with and without M protein to further demonstrate that the behaviour of F and G proteins remain unaltered between those areas.

Furthermore, without some sort of control experiments that put these results into context it is difficult to ascertain the validity of the claim that the incorporation of F and G into budding viruses is not regulated by protein M. Therefore authors could include a control experiments using either externally supplied protein such as GPI-SNAP or one of the cell plasma membrane proteins that do not become incorporated into the virus membrane and compare their behaviour at budding sites with F and G proteins. Moreover, authors should also consider performing control experiments using one of the F protein mutants that was shown by one of the co-authors to alter the level of incorporation of G protein into virus particles (Ref: Johnston et al. 2017 doi:10.1128/JVI.02150-16).

Authors support the conclusion of this section by pointing that there is no difference in the amount of overlap in the F and G proteins in cells with and without M protein (Line 127-128). This is taken as an indication that there are no rearrangements of these proteins inside virus assembly sites. However, since rigorous analysis of the differences was not performed and there are concerns regarding the use of overlapping pixels as an indication of co-association (point 2 of the report) this argument does not appear very strong.

Minor concerns:

- In their result graphs authors utilise Standard Error of the Mean (SEM) to illustrate their error bars. On the other hand t-test statistical analysis used by the authors relies on standard deviation (SD), therefore authors should use SD in their graphs as the use of SEM here is misleading (Ref: Streiner 1996, doi:10.1177/070674379604100805).

- In Supplementary Figure 1 authors should provide a figure legend showing at least basic information on how the SEM image was acquired.

Reviewer #3 (Remarks to the Author):

Q. Liu et al use super-resolution microscopy to study assembly of Nipah virus like particles. A plasmid-based transfection system is applied to examine whether the particle assembly process resembles the general believe that the matrix protein recruits the glycoproteins to budding sites. The authors find that the matrix protein forms dome-like structures that may represent assembly sites, and that the Nipah G and F proteins also localize to these structures, but in a manner that seems rather independent from the matrix protein. Increasing the level of G and F proteins increases the level of G and F incorporation in VLP-like structures at the cell surface. From these data the authors conclude that, rather than recruitment of G and F by matrix protein, that G and F incorporate stochastically in VLPs formed by the matrix protein.

Comments/concerns:

A significant concern for this reviewer is that the VLP assembly condition used (MGF coexpression) may not recapitulate a condition whereby particles assemble optimally. Previous work has shown that VLPs form upon transient expression of Nipah M alone, G and F, or M and G and F. However, it is far from clear whether VLPs form efficiently or just marginally under any of those conditions. In ref Patch et al, 2007, only a minor portion of the expressed proteins appears to assemble into particles. In ref Johnston et al, 2017, the level of MGF-induced budding is only slightly higher than several 2-protein combinations. In the present work, G and F proteins appear to localize/target rather independently from the M protein in several regards. The data presented here are interesting and provide useful information with regard to better understanding of, or optimizing, the generation Nipah VLPs for downstream purposes. However, the data are not sufficient to propose a new virus assembly model as stated in the title, because the data could also reflect suboptimal assembly conditions, in which protein targeting, localization, and interaction are artefactual and not representative of how viral assembly occurs. Can the authors exclude that the M, G, and F proteins are behaving artefactually under the conditions used, due for example to MGF co-expression not allowing efficient assembly, or due to the requirement for another viral protein for optimal VLP assembly, or due to the M protein (or another protein) not being properly post-translationally processed in the absence of replicating virus etc? Whereas interesting and useful observations are made, the system and data are not adequate to support a new virus assembly model. Biosafety conditions will probably not allow a comparison of VLP assembly to virus assembly. However, to support a novel assembly model, more extensive analysis should or could be carried out using the plasmid-based transfection assay to exclude that the observations of protein behavior are artefacts of the system; previous work has not thoroughly examined this.

Some additional comments:

F proteins are highly sensitive in terms of folding and maturation. Reference 20 shows an F protein with a flag tag at the n-term and an AU1 tag at the c-term.

Can the authors exclude that:

- a) the flag tag affects expression and stability of F and indirectly the ability of F to interact with M?
- b) the AU1 tag does not interfere with F-M interaction? Cytoplasmic tail domains of many F proteins (including Hendra F) are important for virus assembly.

Figure 1. The legend states that panel d reflects position 1 of Fig 1A. Is this correct?... looking at the data, panel d seems to reflect position 3.

It would help the reviewer if the authors could add page numbers and use continuous line numbers. It's difficult to keep track of the various sections when reading a printed version.

Reviewer #1 (Remarks to the Author):

In the manuscript by Liu et al., dSTORM was utilized to evaluate the localization of the F, G, and M proteins of Nipah virus, expressed from plasmids. Overall, this is an interesting paper, that comes to the conclusion that the M protein does not effect interactions or incorporations of F and G into VLPs, but that this is stochastic. This is a surprising conclusion, fully supported by the data, but it would be more convincing if a number of experiments were added to the manuscript. It should be noted, that at some point in the arc of this research, these experiments really do need to be performed with the entire virus, to validate the findings. I think this should be mentioned in the manuscript.

Response:

In the revised manuscript, we have pointed out that there are potential differences between the plasmid transfection-based method and virus infection assays, and whether these models are valid in the real virus remains an open question.

Revisions made:

Lines 186-188 on page 7 and 8: “Since NiV is a biosafety level-4 agent, most models on NiV life cycle have been proposed by using plasmid transfection-based methods³¹. Whether these models are valid in the real virus remains an open question.”

Major issues

1) Is it known whether the GFP inserted into the NiV-M, interferes with normal viral processes? Has this been confirmed. GFP addition to other paramyxovirus M proteins impedes filament formation and normal assembly. Given that working with the whole virus is difficult, comparisons should be made with and without the GFP tag, if an anti-M antibody exists, and if not, then with smaller epitope tags, such as Flag or V5 or HA.

Response:

This particular construct was provided by Dr. Benhur Lee at the Icahn School of Medicine at Mount Saini Hospital. Lee’s group has shown that the VLP assembly and budding ability of this NiV-M-GFP construct was comparable to the untagged NiV-M (reference 30: Wang et al., 2010).

Revisions made:

Lines 201-202 on page 8: “The VLP assembly and budding ability of this NiV-M-GFP construct was comparable to the untagged NiV-M³⁰.”

2) Based on some of their recent work (Johnston et al., 2017) – it would be very useful to image G and M and F, G, M in order to see if F facilitates F/M colocalization. In the recent paper in JVI, they demonstrate that F facilitates G incorporation into VLPs, therefore seeing if F effects G localization with M structures would make sense.

Response:

We thank the reviewer's suggestions to improve our manuscript. To study whether F affects G's co-localization with M clusters, we have carried out SMLM imaging of G and M in the absence and presence of F (Extended Data Fig. 5c and d). The results show segregated G and M nanoclusters on the plasma membrane in both cases. Additionally, we have carried out a numerical co-localization analysis. The analysis shows a negative averaged Degree of co-localization (DoC) value between F and M with and without F (Extended Data Fig. 5e) This observation suggests a noncorrelated distribution between G and M regardless of the presence of F. Furthermore, the dual-color SMLM images and DoC analysis of M and F (Extended Data Fig 5a and b) and F and G (Fig. 2e and f) also indicate noncorrelated distributions among these three molecules on the plasma membrane.

Revisions made:

- (a) Lines 87-99 on page 3: "A numerical co-localization analysis was carried out using the coordinate-based co-localization (CBC) algorithm previously developed by Malkusch et al²⁶. The degree of co-localization (DoC) is calculated for every single-molecule localization and has a value from -1 for segregation, through 0 for random distributions, to 1 for a high probability of correlated co-localization²⁶⁻²⁸. A random distribution indicates that the co-localization of two molecules results from a random event rather than a specific mechanism. In this algorithm, the DoC value is dependent on the maxima radius (R_{max}) used for the DoC analysis^{26,27}. A R_{max} of 100 nm was used in this study to reflect the size of a typical VLP. The effect of the selected R_{max} on the DoC values is demonstrated in the Extended Data Fig. 3. The analysis indicates that F and G are mostly localized in segregated clusters as the DoC values show a maximum in the negative range (Fig. 2f). The distribution of the DoC values suggests that there is no specific mechanism for correlated co-localization between F and G. However, it does not exclude the possibility of co-localization between F and G as random events. The analysis partially explain that the association of the glycoproteins observed by co-immunoprecipitation assays can be due to these random events^{22,29,30},"
- (b) Lines 136-138: "Previous studies using Western Blot analysis suggest that F may facilitate G to incorporate into VLPs⁹. Nonetheless, we did not observe a significant difference on the DoC values between M and G with or without the presence of F (Extended Data Fig.5c, d and e)."
- (c) Lines 132-134: "Furthermore, dual-color SMLM images indicate noncorrelated distributions among M, F, and G with negative averaged DoC values (Fig. 2f and Extended Data Fig.5)."

3) If the F-cytoplasmic tail is removed - does it change the associations?

Response:

As described above, our new data show noncorrelated distributions among F, G, and M. Therefore, removing the F-cytoplasmic tail is unlikely to change the associations.

- 4) Is M localized to caveolae? Are F and G in different membrane structures?
- 5) If you remove cholesterol - does this alter the localization of M?
- 6) Where is the actin cortex in these images? Does the cortex deform during M expression? Where are the microtubules relative to these structures?
- 7) Endothelial cells, one of the targets of NiV, are polarized. Does polarization effect the localization of the F, G or M? MDCKs (Lamp et al., JVI, 2013) on transwells could also be used.

Response:

These are important questions about viral-host interactions during the virus assembly and budding processes, but they are beyond the scope of the current study. Our major conclusion that M does not actively recruit F and G to the nascent assembly sites remains valid regardless of the mechanisms of the viral-host interactions.

Reviewer #2 (Remarks to the Author):

In this study, authors utilise the super-resolution single molecules localisation microscopy (SMLM) approach to investigate the behaviour of Nipah virus surface glycoproteins G and F (responsible for virus attachment and fusion, respectively) at virus assembly sites. Here, authors conclude that the incorporation of the virus surface proteins into newly assembled virus proceeds in stochastic manner that is not modulated by the viral structural protein M. This virus assembly model differs to the virus assembly models proposed in previous studies of viruses such as HIV-1 or influenza where incorporation of virus surface proteins is thought to be dependent on direct or indirect interactions with virus structural proteins.

In general, the manuscript is short, concise and its story is easy to follow and understand. On the other hand, this study suffers from issues which cast doubts on authors' claims and therefore need to be addressed.

Concerns:

1. In this study authors utilise full antibodies to tag the viral proteins of interest. The multivalent nature of both primary and secondary antibodies used raises concerns that it will induce artificial clustering of the proteins. Such clustering may therefore give a false result with regards to the observed distribution of viral proteins on the virus surface, especially in single molecule resolution SMLM approach. This is especially a concern in the samples that were post-fixed following the antibody labelling (Figure 2 and 3). Furthermore, since even a very long and strong fixation does not complexly immobilise the proteins (Ref: Tanaka et al. 2010 doi:10.1038/nmeth.f.314) antibody-induced protein clustering may also be a problem in pre-fixed samples (Figure 1).

To address this issue authors should utilise Fab fragments for both primary and secondary antibodies for their imaging experiments. At the very least authors should conduct Fab based control experiment that assesses the potential impact of antibody induced protein clustering on the distribution of investigated proteins.

Response:

- (1) As described in the figure legends, all samples were pre-fixed before immunostaining, not post-fixed in this study.
- (2) Following the reviewer's suggestions, we have carried out experiments using Fab fragments. However, the labeling density of the Fab fragments was too low for us to identify the dorsal surface of the cell. As described in the manuscript, the dome-like structures formed by M during assembly and budding (Fig.1e-f; 3b and c) can only be observed at the dorsal surface, not the ventral surface of the cell in contact with the coverslip. Locating the dorsal surface using single-molecule localization microscopy is a difficult practice because the majority of the fluorophores are switched to the dark state. A high labeling density is needed to visualize the dorsal surface. The full antibody used in our study is currently the best option to study cellular structures at the dorsal surface.
- (3) We fixed cells with 4% PFA and 0.2% GA for 90 min (Tanaka et al. 2010). NiV-F and -G are transmembrane proteins. According to Tanaka's study, the percentage of mobile molecules of a transmembrane protein (TfR) on the plasma membrane using this fixation protocol is less than 5%. Therefore, the antibody-induced clustering effect is negligible in our study.

2. In Figure 2 authors claim that lack of significant overlap between proteins F and G, signified by very few yellow "overlap" pixels (Line 88), indicate that these proteins have a limited association (Line 90) in contrast to previous bulk studies (Line 91). However, given the fact that authors themselves state (line 89) that the localization precision of the microscope used is ~10 nm which is roughly the size of a single antibody the co-localisation estimation by yellow "overlap" pixels can be misleading. At such high accuracy and precision levels the size of tagging antibodies themselves will prevent any overlap as no two antibody complexes can occupy the same physical space.

Authors should indicate this caveat more clearly or use another method to assess the protein co-association that have been used in super-resolution SMLM experiments (Ref: Nicovich et al. 2017 doi:10.1038/nprot.2016.166 and Georgieva et al. 2016 doi:10.1016/j.ymeth.2016.03.029). Furthermore, the use of the relatively large full antibody tags may also cause selective lack of labelling of very closely associated F and G proteins due to the occlusion by the antibody complexes themselves. Such behaviour may result in the perceived lack of association between proteins. This would normally not be a problem in a classic diffraction limited microscopy experiments but becomes a potential pitfall in super-resolution based studies of protein association.

To address this authors should consider performing an experiment with smaller tags such as Fabs or nanobodies.

Response:

We thank the reviewer's suggestions to improve our manuscript. Following the suggestion by the reviewer, the degree of co-localization (DoC) analysis using the coordinate-based co-localization (CBC) algorithm described by Malkusch et al. and Georgieva et al. has been carried out in the revised paper. The analysis shows noncorrelated distributions among M, F, and G with negative averaged DoC values (Fig. 2f and Extended Data Fig.5). These results are consistent with our proposed model.

Revisions made:

(1) Lines 87-99 on page 3: "A numerical co-localization analysis was carried out using the coordinate-based co-localization (CBC) algorithm previously developed by Malkusch et al.²⁶. The degree of co-localization (DoC) is calculated for every single-molecule localization and has a value from -1 for segregation, through 0 for random distributions, to 1 for a high probability of correlated co-localization²⁶⁻²⁸. A random distribution indicates that the co-localization of two molecules results from a random event rather than a specific mechanism. In this algorithm, the DoC value is dependent on the maxima radius (R_{max}) used for the DoC analysis^{26,27}. A R_{max} of 100 nm was used in this study to reflect the size of a typical VLP. The effect of the selected R_{max} on the DoC values is demonstrated in the Extended Data Fig. 3. The analysis indicates that F and G are mostly localized in segregated clusters as the DoC values show a maximum in the negative range (Fig. 2f). The distribution of the DoC values suggests that there is no specific mechanism for correlated co-localization between F and G. However, it does not exclude the possibility of co-localization between F and G as random events. The analysis partially explain that the association of the glycoproteins observed by co-immunoprecipitation assays can be due to these random events^{22,29,30}"

(2) Additional DoC analyses in Fig. 2f, 3g, Extended Data Fig. 3, and Extended Data Fig. 5b and e.

3. In Figure 3, which provides the data for the main point of this study, authors conclude that since there is no difference in the localization densities of F and G proteins in the areas with or without M protein this indicates that the incorporation of these proteins to the budding viruses may proceed in purely stochastic without any interactions with protein M.

With such a strong claim which represents the main conclusion of the paper, it is surprising that the authors did not perform a more in-depth analysis of the data to strengthen their case, especially since it is more difficult to convincingly prove a lack of a process than to demonstrate that it exists. For example, as with Figure 2 data, authors should consider performing Hopkins index analysis (or other spatial analysis) of F and G protein distributions in the areas with and without M protein to further demonstrate that the behaviour of F and G proteins remain unaltered between those areas.

Response:

Following the suggestion by the reviewer, Hopkins' index analysis and the DoC analysis have been carried out in the revised paper (Fig. 3f and g).

Revisions made:

Lines 131-132 on page 5: “Additionally, the Hopkins’ indices (Fig. 3f) and DoC values (Fig. 3g) of F and G were comparable in the M-positive and M-negative regions.”

Furthermore, without some sort of control experiments that put these results into context it is difficult to ascertain the validity of the claim that the incorporation of F and G into budding viruses is not regulated by protein M. Therefore authors could include a control experiments using either externally supplied protein such as GPI-SNAP or one of the cell plasma membrane proteins that do not become incorporated into the virus membrane and compare their behaviour at budding sites with F and G proteins. Moreover, authors should also consider performing control experiments using one of the F protein mutants that was shown by one of the co-authors to alter the level of incorporation of G protein into virus particles (Ref: Johnston et al. 2017 doi:10.1128/JVI.02150-16).

Response:

- (1) Whether VLPs would incorporate other surface proteins in the host cell is beyond the scope of the current study. The outcomes of these experiments will not change our conclusion that NiV-M does not actively recruit NiV-F or G during assembly.
- (2) To study whether F affects G’s co-localization with M clusters, we have carried out SMLM imaging of G and M in the absence and presence of F (Extended Data Fig. 5c and d). The results show segregated G and M nanoclusters on the plasma membrane in both cases. Additionally, we have carried out a numerical co-localization analysis. The analysis shows a negative averaged Degree of co-localization (DoC) value between F and M with and without F (Extended Data Fig. 5e) This observation suggests a noncorrelated distribution between G and M regardless of the presence of F. Furthermore, the dual-color SMLM images and DoC analysis of M and F (Extended Data Fig 5a and b) and F and G (Fig. 2e and f) also indicate noncorrelated distributions among these three molecules on the plasma membrane. F protein mutants are unlikely to change these noncorrelated distribution patterns.

Revisions made:

- (a) Lines 136-138: “Previous studies using Western Blot analysis suggest that F may facilitate G to incorporate into VLPs⁹. Nonetheless, we did not observe a significant difference on the DoC values between M and G with or without the presence of F (Extended Data Fig.5c-e).”
- (b) Lines 132-134: “Furthermore, dual-color SMLM images indicate noncorrelated distributions among M, F, and G with negative averaged DoC values (Fig. 2f and Extended Data Fig.5).”

Authors support the conclusion of this section by pointing that there is no difference in the amount of overlap in the F and G proteins in cells with and without M protein (Line 127-128). This is taken as an indication that there are no rearrangements of these proteins inside virus assembly sites. However, since rigorous analysis of the differences was not performed and there are concerns regarding the use of overlapping pixels as an indication of co-association

(point 2 of the report) this argument does not appear very strong.

Response:

Following the suggestion by the reviewer, Hopkins' index analysis and a coordinate-based colocalization analysis have been carried out in the revised paper (Fig. 3f and g), as described above in our response to the reviewer's comment 1.

Revisions made:

Lines 131-132 on page 5: "Additionally, the Hopkins' indices (Fig. 3f) and DoC values (Fig. 3g) of F and G were comparable in the M-positive and M-negative regions."

Minor concerns:

- In their result graphs authors utilise Standard Error of the Mean (SEM) to illustrate their error bars. On the other hand t-test statistical analysis used by the authors relies on standard deviation (SD), therefore authors should use SD in their graphs as the use of SEM here is misleading (Ref: Streiner 1996, doi:10.1177/070674379604100805).

Revisions made:

We have followed the reviewer's suggestion and used mean and standard deviation (SD) in Fig. 2, Fig. 3, and Extended Data Fig. 2 and Extended Data Fig. 4.

- In Supplementary Figure 1 authors should provide a figure legend showing at least basic information on how the SEM image was acquired.

Revision made:

The figure legend is added.

Reviewer #3 (Remarks to the Author):

Q. Liu et al use super-resolution microscopy to study assembly of Nipah virus like particles. A plasmid-based transfection system is applied to examine whether the particle assembly process resembles the general believe that the matrix protein recruits the glycoproteins to budding sites. The authors find that the matrix protein forms dome-like structures that may represent assembly sites, and that the Nipah G and F proteins also localize to these structures, but in a manner that seems rather independent from the matrix protein. Increasing the level of G and F proteins increases the level of G and F incorporation in VLP-like structures at the cell surface. From these data the authors conclude that, rather than recruitment of G and F by matrix

protein, that G and F incorporate stochastically in VLPs formed by the matrix protein.

Comments/concerns:

A significant concern for this reviewer is that the VLP assembly condition used (MGF coexpression) may not recapitulate a condition whereby particles assemble optimally. Previous work has shown that VLPs form upon transient expression of Nipah M alone, G and F, or M and G and F. However, it is far from clear whether VLPs form efficiently or just marginally under any of those conditions. In ref Patch et al, 2007, only a minor portion of the expressed proteins appears to assemble into particles. In ref Johnston et al, 2017, the level of MGF-induced budding is only slightly higher than several 2-protein combinations. In the present work, G and F proteins appear to localize/target rather independently from the M protein in several regards. The data presented here are interesting and provide useful information with regard to better understanding of, or optimizing, the generation Nipah VLPs for downstream purposes. However, the data are not sufficient to propose a new virus assembly model as stated in the title, because the data could also reflect suboptimal assembly conditions, in which protein targeting, localization, and interaction are artefactual and not representative of how viral assembly occurs. Can the authors exclude that the M, G, and F proteins are behaving artefactually under the conditions used, due for example to MGF co-expression not allowing efficient assembly, or due to the requirement for another viral protein for optimal VLP assembly, or due to the M protein (or another protein) not being properly post-translationally processed in the absence of replicating virus etc? Whereas interesting and useful observations are made, the system and data are not adequate to support a new virus assembly model. Biosafety conditions will probably not allow a comparison of VLP assembly to virus assembly. However, to support a novel assembly model, more extensive analysis should or could be carried out using the plasmid-based transfection assay to exclude that the observations of protein behavior are artefacts of the system; previous work work has not thoroughly examined this.

Response:

We appreciated the reviewer's comments. The reviewer summarized many potential issues associated with the plasmid-based transfection assay, which are commonly used in research on biosafety level 4 agents. As stated by the reviewer, biosafety conditions make it very difficult to study level-4 virus assembly. The commonly-believed assembly model, challenged by our findings, was also established using plasmid-transfection assays rather than virus infection assays. Therefore, our study reports a fair comparison. We are aware of the difference between the assembly of VLPs and real viruses, and we have made this clear in the revised manuscript.

Revisions made:

Lines 186-188 on page 7: "Since NiV is a biosafety level-4 agent, most models on NiV life cycle have been proposed by using plasmid transfection-based methods³¹. Whether these models are valid in the real virus remains an open question."

Some additional comments:

F proteins are highly sensitive in terms of folding and maturation. Reference 20 shows an F protein with a flag tag at the n-term and an AU1 tag at the c-term.

Can the authors exclude that:

a) the flag tag affects expression and stability of F and indirectly the ability of F to interact with M?

Response:

This construct was published by Stone et al. (reference 20, Stone et al., 2016). The FLAG-tagged F construct is fully functional in a cell-cell fusion assay.

Revisions made:

Lines 203-204 on page 8: "The FLAG-tagged F is fully functional in a cell-cell fusion assay"²⁰

b) the AU1 tag does not interfere with F-M interaction? Cytoplasmic tail domains of many F proteins (including Hendra F) are important for virus assembly.

Response:

We did not use an AU1-tagged F construct. As described in the Materials and Methods (lines 203-204), we used a FLAG-tagged NiV-F construct.

As shown in Fig 2f, 3g, and Extended data fig 5, our data suggest noncorrelated distributions among F, G, and M. Therefore, removing the F-cytoplasmic tail is unlikely to change the associations.

Figure 1. The legend states that panel d reflects position 1 of Fig 1A. Is this correct?... looking at the data, panel d seems to reflect position 3.

Response:

The typo has been corrected. It should be position 3.

It would help the reviewer if the authors could add page numbers and use continuous line numbers. It's difficult to keep track of the various sections when reading a printed version.

Response:

We have added page numbers and used continuous line numbers in the revised manuscript.

Reviewers' comments:

Reviewer #1 (Remarks to the Author):

Overall, I'm happy with the responses. The colocalization analysis was well done and supports the conclusion.

The question is whether this is a cell specific phenomena and whether this is different during a true viral infection. You addressed the issue of the actual infection, but cell phenotype was not addressed.

I do think that polarized cells may make a difference here. An experiment using a cell line that polarizes, even on glass, would be useful for validating the data. Lamp et al., JVI, 2013, demonstrated that NiV is released from the apical side of polarized cells, and that apical trafficking was important. Could that mechanism alter your findings? This is not entirely necessary for publication, but would greatly strengthen your findings.

Reviewer #2 (Remarks to the Author):

The authors have satisfactorily addressed my concerns and queries brought up in the previous assessment. I am happy to support this manuscript for publication.

A few minor points:

- In Methods section authors should include the information on what software was used to reconstruct SMLM images
- Fig 3f – Y-axis should read “%Normalized Hopkins’ index”
- Line 216 – misspelled “fluorescence”

Reviewer #3 (Remarks to the Author):

Liu et al have used super-resolution microscopy to study assembly of Nipah virus like particles, based on transient expression of the NiV M, G and F proteins. Relative to the previous submission, additional SMLM imaging data and co-localization analysis were added to support the claim of a novel stochastic assembly model, and some of the concerns have been adequately addressed. Two significant concerns remain:

In regards to flag-tagged F and its functionality: In response to the previous review, authors state that their F construct (from Stone et al 2016, ref 21) contains only a flag tag and that it is fully (fusion) functional.

However, the paper referenced by the authors (Stone et al) shows in Fig 1B that the flag-tagged F protein contains both a flag tag at the cleavage site and an AU1 tag at the very c-term. If the Stone paper is in error, then this needs to be rectified. In addition, the authors state that the flag-tagged F is fully functional in a cell-cell fusion assay. It appears that this

has not been examined, as Stone et al report that a fusion assay (Fig 4E) was done with wt F. In support of this, flag-tagged F has the tag located right at the cleavage site adjacent to the fusion peptide, which makes it unlikely that it remains fusion-competent. Due to the importance of F for the outcome of this paper, the authors need to clarify and describe in their manuscript what the flag-tagged F construct is. From the referenced Stone et al, one concludes it contains the AU1 tag at the c-term as well. If the c-term plays a role in interaction with M, which is the case for many other paramyxoviruses, then the AU1 tag at that location could result in exactly what the authors report: a stochastic incorporation, potentially due however to the M-F interaction being interrupted.

With regard to claim of a stochastic model:

It is acceptable that VLP studies are used in place of viral studies given the BSL-4 status of NiV. And this study and other studies provide useful information. The issue above needs to be clarified to exclude potential impact of interrupted F-M interaction, before any claims such as in the title can be made.

Even if the F used is not the F shown in Fig 1B of Stone et al 2016, and does not contain a c-terminal tag, the evidence for the claim made in the title is not yet convincing, in spite of the added data. The concern that MGF co-expression may create particle assembly conditions that are too artefactual or suboptimal to support the strong claim in the title (stated in the previous review), has not been addressed. Any factor that is missing, or post-translational modification that does not occur under the conditions used, could render assembly suboptimal and create unrealistic and potentially stochastic conditions. Either additional analyses of co-expression conditions are needed to increase the likelihood that the conditions used recapitulate viral particle assembly and thereby support the title, or claims need to be moderated.

Reviewer #1 (Remarks to the Author):

Overall, I'm happy with the responses. The colocalization analysis was well done and supports the conclusion.

The question is whether this is a cell specific phenomena and whether this is different during a true viral infection. You addressed the issue of the actual infection, but cell phenotype was not addressed.

I do think that polarized cells may make a difference here. An experiment using a cell line that polarizes, even on glass, would be useful for validating the data. Lamp et al., JVI, 2013, demonstrated that NiV is released from the apical side of polarized cells, and that apical trafficking was important. Could that mechanism alter your findings? This is not entirely necessary for publication, but would greatly strengthen your findings.

Response:

We thank the reviewer for the positive comments and suggestions. The effects of cell phenotype and polarization will be considered for future studies.

Reviewer #2 (Remarks to the Author):

The authors have satisfactorily addressed my concerns and queries brought up in the previous assessment. I am happy to support this manuscript for publication.

A few minor points:

- In Methods section authors should include the information on what software was used to reconstruct SMLM images
- Fig 3f – Y-axis should read “%Normalized Hopkins’ index”
- Line 216 – misspelled “fluorescence”

Response:

We thank the reviewer for the positive comments. The following revisions have been made.

(1) Custom software written in MATLAB (Mathworks) was used to reconstruct SMLM images (See the revised Materials and Methods.)

(2) The typos were corrected.

Reviewer #3 (Remarks to the Author):

Liu et al have used super-resolution microscopy to study assembly of Nipah virus like particles, based on transient expression of the NiV M, G and F proteins. Relative to the previous submission, additional SMLM imaging data and co-localization analysis were added to support the claim of a novel stochastic assembly model, and some of the concerns have been adequately addressed. Two significant concerns remain:

In regards to flag-tagged F and its functionality: In response to the previous review, authors state that their F construct (from Stone et al 2016, ref 21) contains only a flag tag and that it is fully (fusion) functional.

However, the paper referenced by the authors (Stone et al) shows in Fig 1B that the flag-tagged F protein contains both a flag tag at the cleavage site and an AU1 tag at the very c-term. If the Stone paper is in error, then this needs to be rectified. In addition, the authors state that the flag-tagged F is fully functional in a cell-cell fusion assay. It appears that this has not been examined, as Stone et al report that a fusion assay (Fig 4E) was done with wt F. In support of this, flag-tagged F has the tag located right at the cleavage site adjacent to the fusion peptide, which makes it unlikely that it remains fusion-competent. Due to the importance of F for the outcome of this paper, the authors need to clarify and describe in their manuscript what the flag-tagged F construct is. From the referenced Stone et al, one concludes it contains the AU1 tag at the c-term as well. If the c-term plays a role in interaction with M, which is the case for many other paramyxoviruses, then the AU1 tag at that location could result in exactly what the authors report: a stochastic incorporation, potentially due however to the M-F interaction being interrupted.

Response:

- (1) The FLAG-tagged NiV-F construct used in this study contains a FLAG tag at the C-terminus of the F2 subunit, and it does not have a C-terminal AU1 tag. The diagram of the construct is shown in Extended Data Fig 1a.
- (2) The FLAG-tagged NiV-F construct is functional in cell-cell fusion assays of three cell lines. Results for 293T cells are shown in Extended Data Fig 1b-d.

With regard to claim of a stochastic model:

It is acceptable that VLP studies are used in place of viral studies given the BSL-4 status of NiV. And this study and other studies provide useful information. The issue above needs to be clarified to exclude potential impact of interrupted F-M interaction, before any claims such as in the title can be made. Even if the F used is not the F shown in Fig 1B of Stone et al 2016, and does not contain a c-terminal tag, the evidence for the claim made in the title is not yet convincing, in spite of the added data. The concern that MGF co-expression may create particle assembly conditions that are too artefactual or suboptimal to support the strong claim in the title (stated in the previous review), has not been addressed. Any factor that is missing, or post-translational modification that does not occur under the conditions used, could render assembly suboptimal and create unrealistic and potentially stochastic conditions. Either additional analyses of co-expression conditions are needed to increase the likelihood that the conditions used recapitulate viral particle assembly and thereby support the title, or claims need to be moderated.

Response:

We appreciate the reviewer's comments and agreement on the difficulties in studying the level-4 virus.

(1) As described above, the F used does not contain a c-terminal tag. It should not interrupt the F-M interaction proposed in previous studies using conventional methods, such as western blot or diffraction-limited microscopy.

(2) We are willing to revise the title although we think that our current title adequately describes our work. "A model" is a relative weak claim. A scientific model presents a set of ideas for phenomena observed experimentally. When we call it a model, it already means that it is tentative. Different models are used to describe our preliminary understanding before a complete theory is developed. A scientific theory is then a broad explanation for how nature works that is widely accepted as true. Here we are proposing "a model", not "the theory", for virus assembly.

Before a complete theory is developed, new models are often proposed when new technologies are available. In the past two years, we have collected hundreds of super-resolution images of these viral proteins but we have not been able to find a single image to support the existing model. Here, we are proposing a possible model previously unrecognized using traditional techniques. As technology continues to advance, we do not know whether our proposed model will be consistent with the ultimate theory for virus assembly. However, we think researchers in this field should be aware that direct visualization of these viral proteins on the host cell's membrane tells a new story about the assembly process.

If the reviewer would insist that our title is too strong and should be moderated, we are willing to moderate our title to "A Possible Virus Assembly Model Revealed by Super-Resolution Microscopy", "New Evidence for Viral Protein Assembly on Cell Membrane Revealed by Super-Resolution Microscopy", or something else suggested by the reviewer. However, it does not fundamentally change what our data show us. Our data have left us no choice but to propose a new model. Since all our data suggest the assembly is stochastic, our current title is pretty informative to our readers. Just like other models, it remains to be tested before the complete theory is developed.

REVIEWERS' COMMENTS:

Reviewer #3 (Remarks to the Author):

Liu et al have used super-resolution microscopy to study assembly of Nipah virus like particles, based on transient expression of the NiV M, G and F proteins. Relative to the previous submission, additional data regarding structure/function of the tagged F protein were provided to clarify which F construct was used for their studies, and an argument in support of their model and title was provided. Although the possibility remains that the plasmid based system does not accurately reflect how Nipah virus assembles, the argument made is reasonable.

Altogether, the data are well done, well presented, and interesting. No further concerns remain.

REVIEWERS' COMMENTS:

Reviewer #3 (Remarks to the Author):

Liu et al have used super-resolution microscopy to study assembly of Nipah virus like particles, based on transient expression of the NiV M, G and F proteins. Relative to the previous submission, additional data regarding structure/function of the tagged F protein were provided to clarify which F construct was used for their studies, and an argument in support of their model and title was provided. Although the possibility remains that the plasmid based system does not accurately reflect how Nipah virus assembles, the argument made is reasonable.

Altogether, the data are well done, well presented, and interesting. No further concerns remain.

Response:

We thank the reviewer for the positive comments.